# WorkflowAgent: Towards Specialized Web Agents Using Production-Scale Workflow Data

## Abstract

LLM agents are rapidly improving to handle increasingly complex web-based tasks. Most of these agents rely on general-purpose, proprietary models like GPT-4 and focus on designing better prompts to improve their planning abilities. However, general-purpose LLMs are not specifically trained to understand specialized web contexts such as HTML, and they often struggle with long-horizon planning. We explore an alternative approach that fine-tunes open-source LLMs using production-scale workflow data collected from over 250 domains corresponding to 6 billion tokens. This approach shows substantial gains over prompting-based agents on existing benchmarks—our agent achieves state-of-the-art direct generation performance on Mind2Web and improves the task success rate by 7.3% over the previous best text-only web agents on WebArena. We perform detailed ablation studies on various fine-tuning design choices and provide valuable insights into LLM selection, training recipes, context window optimization, and effect of dataset sizes.

## 1 Introduction

Large language model (LLM) agents have advanced significantly in web navigation. They can carry out user-specified tasks by reasoning what actions to take and what external resources to interface with. Recent studies (Zheng et al., 2024; Lai et al., 2024; Zhang et al., 2024; Song et al., 2024) have shown that, with better planning and exploration strategies, LLM agents can independently solve various web tasks, such as locating a specific Wikipedia page and booking flights or restaurants.

Despite these improvements, the performance of existing agents on complex web navigation benchmarks remains significantly below human levels (Deng et al., 2023; Zhou et al., 2024; Drouin et al., 2024). One possible reason is that they depend heavily on *general-purpose* LLMs. Indeed, all top-performing agents like WebPilot (Zhang et al., 2024), AWM (Wang et al., 2024b), and SteP (Sodhi et al., 2024) rely on prompting proprietary models like GPT-4 (OpenAI, 2024a). These general-purpose LLMs are not optimized for interpreting web contexts such as HTML or accessibility trees; their pre-training and alignment processes do not address navigation-related challenges; and their proprietary nature presents a major obstacle in adapting them to web environments via post-training.

In this work, we explore an alternative approach by fine-tuning open-source LLMs with a large set of real-world workflow data[1] to develop *specialized* web agents (Figure 1). Through extensive experiments, we show that this approach not only boosts the web understanding and planning abilities of LLMs, achieving state-of-the-art results on various benchmarks, but also allows us to develop agent models significantly smaller than proprietary LLMs, reducing the serving costs. Our empirical results suggest that large-scale, high-quality, real-world data can be essential to agent development.

Specifically, leveraging an AI workflow documentation software (name omitted for double-blind review), we collected a large set of workflow data representing action sequences executed by real users in real web environments. This dataset encompasses a diverse spectrum of websites (over 250 domains and 10,000

---

[1]Due to privacy concerns, we restrict access to our collected data. However, we will release the preprocessing, training, inference code, and a copilot (i.e., browser extension) powered by our agent.

Figure 1: **Left:** Most existing LLM web agents are built on top of general-purpose, proprietary models like GPT-4 and rely heavily on prompt engineering. Their performance is enhanced by leveraging external planning, reasoning, and memory modules. **Right:** We explore an alternative way to develop specialized agents by fine-tuning open-source LLMs using high-quality, real-world workflow data. This significantly boosts agent's planning capacity, enabling it to outperform proprietary models with small LLMs that have lower serving costs.

subdomains), task objectives, task difficulty, and task length. Each step in the workflow features not only the raw HTML-DOM of the website but also a comprehensive documentation of the action, including natural language description, mouse or keyboard operation, and the CSS selector of the target element. We reformat the data into a next-step prediction formulation and use them to fine-tune open-source LLMs.

With access to this production-scale dataset, we develop WorkflowAgent, the first family of specialized, single-stage LLM agents capable of directly generating the next step based on the website's DOM and action history. This is in contrast with previous fine-tuned agents that require multiple stages to produce an action, e.g., first narrowing down to a set of candidate elements and then selecting one from the candidates (Deng et al., 2023). To evaluate the capacity and generalization ability of WorkflowAgent, we test it on public benchmarks without any further training. WorkflowAgent significantly outperforms existing GPT-4 and multi-stage agents. Notably, the 32B WorkflowAgent-Large achieves state-of-the-art direct generation performance on Mind2Web (Deng et al., 2023), with step success rate surpassing the baselines by 5-10% across all test sets. On the interactive benchmark WebArena (Zhou et al., 2024), our 7B WorkflowAgent-Small improves the previous best task success rate from 45.7% to 51.3%. WorkflowAgent-Large achieves an even better task success rate of 53%, marking the highest performance among text-only LLM agents.

Beyond the empirical results, our work provides several insights valuable for future research: (1) we show that direct fine-tuning on HTML is feasible and can improve the agent's ability in identifying the correct target; (2) we design an effective HTML preprocessing pipeline that balances between preserving essential information and minimizing context length; (3) we provide a thorough analysis on various design choices in fine-tuning, such as LLM backbone and context window selection; (4) we illustrate how fine-tuning improves agent performance as dataset size increases.

Our work highlights the potential of building strong web agents via specialized fine-tuning. This approach not only improves the agent's planning abilities relative to prompt-engineered alternatives, but also reduces inference costs due to the smaller sizes of open-source LLMs. While our work focuses on fine-tuning, it can be extended to leverage search or memory modules (Koh et al., 2024; Wang et al., 2024b), combined with existing planning frameworks (Yao et al., 2022; Madaan et al., 2023; Shinn et al., 2023), or integrated into multi-modal agent systems (Wang et al., 2024a; Shen et al., 2025; Liang et al., 2025). We view WorkflowAgent as an important step towards developing fully automated agents for real-world web applications.

## 2 Related Work

**Prompting-based agent frameworks.** The majority of web agent works reuse existing LLMs and propose different prompting strategies to improve action prediction. One line of research focuses on exploiting previous experience via self-feedback (Sun et al., 2023; Shen & Yang, 2021; Yuan et al., 2021a;b) or in-context demonstrations (Fu et al., 2024; Zheng et al., 2024; Wang et al., 2024b; Ou et al., 2024; Shen et al., 2024b). A separate line of work centers around encouraging exploration by including external evaluators (Pan et al., 2024), using synthesized instructions (Murty et al., 2024), or applying more sophisticated search algorithms

like stack (Sodhi et al., 2024), best-first tree search (Koh et al., 2024), or Monte Carlo Tree Search (Zhang et al., 2024). Despite the research efforts, these prompting methods rely heavily on the quality of the LLM used. Open-source models such as LLaMA (Dubey et al., 2024), Code LLaMA (Rozière et al., 2024), and Flan-T5 (Chung et al., 2022) generally underperform proprietary models like GPT-4. However, fine-tuning proprietary LLMs can often be costly and challenging, as it is restricted to being done through APIs. This implies an opportunity for enhancing open-source LLMs to match or outperform proprietary agents.

**Fine-tuning-based web agents.** Compared to developing better prompting frameworks, comparatively less attention has been given to optimizing the LLMs themselves to better handle web environments (Xu et al., 2024). Due to the difficulty of directly generating a single target element from the raw HTML, which often contains thousands of elements, existing works such as MindAct (Deng et al., 2023) leverage a two-stage pipeline, which first uses a small fine-tuned LM to rank the web elements and then a large fine-tuned LM to select from the top elements. WebAgent (Gur et al., 2023) fine-tunes a 540B Flan-UPalm to generate code for controlling web pages. More recently, AutoWebGLM (Lai et al., 2024) and WebRL (Qi et al., 2025) train agent models using a combination of curriculum learning, reinforcement learning, and rejection sampling fine-tuning. Despite the complicated training procedures, these methods often underperform GPT-4-based prompting agents. In contrast, our work shows that fine-tuning can outperform GPT-4 and even o1-preview (OpenAI, 2024b). We also note that OpenAI's Operator (OpenAI, 2025) was released after we had completed this work, which is why it was not included as an evaluation baseline. Nonetheless, our text-only agent underperforms the multi-modal Operator only by a small margin.

Beyond the aforementioned work, there is an earlier line of research that fine-tunes LLMs for HTML inputs (Gur et al., 2022; Nakano et al., 2022; Liu et al., 2023). However, their primary application is question-answering tasks, such as answering "could sunflowers really track the sun across the sky", and they cannot be used to generate a sequence of actions based solely on the user objective.

Lastly, an emerging line of research including CogAgent (Hong et al., 2023), SeeClick (Cheng et al., 2024), WebGUM (Furuta et al., 2024), and WebVoyager (He et al., 2024) develops multi-modal web agents using screenshots. However, our current version of WorkflowAgent focuses exclusively on text inputs due to a lack of accompanying visual data. Thus, we do not compare with multi-modal methods in our experiments and leave developing multi-modal WorkflowAgent as future work.

## 3 Method

In this section, we first overview the general setup of solving web-based tasks with LLM agents. Then, we detail our proposed method to develop specialized agents from open-source LLMs.

### 3.1 General Setup

We consider solving a web-based task as a sequential decision-making process guided by a high-level objective. For each task, the user specifies an objective and a starting web page. Then, at every step, the agent outputs an action based on the task objective, the current web page, and the history. Formally, denote the user objective as $q$. The web environment is governed by a transition function $T$ that can evolve over time. The agent is instantiated by a language model $L$. At each time step $t$, the agent observes $o_t$ produced by the environment state $s_t$ and history $h_t = H(o_{1:t-1}, a_{1:t-1})$. It outputs an action $a_t = L(q, o_t, h_t)$, which is executed in the environment, and the state changes correspondingly $s_{t+1} = T(s_t, a_t)$. This iterative process stops when the agent issues a stop signal, or a task termination condition is met, such as reaching a predefined maximum number of steps.

For single-modal, text-only agents, the observation $o_t$ typically consists of the website's URL, the HTML-DOM (Object Model for HTML, which defines HTML elements and their properties, methods, and events), and the accessibility tree (a representation that can be understood by assistive technologies like screen readers). Since the raw HTML-DOM is often long and contains redundant structural information, most methods employ pruning strategies, which could be as simple as retaining a fixed set of HTML tags and attributes or more complex ones like LLM-based element ranking and filtering (Deng et al., 2023).

The action $a_t$ emulates the keyboard and mouse operations available on web pages. The most general action space in existing work consists of element operations, such as clicking, typing, and key combination pressing; tab actions, such as opening, closing, and switching between tabs; navigation actions, such as going forward and backward in the browsing history (Zhou et al., 2024).

As discussed earlier, previous web agent work focuses on presenting useful demonstrations through $h_t$ or iteratively revising $a_t$ to improve the quality of the predicted next step. In contrast, we explore whether we can improve the model $L$ itself by learning from a vast amount of data and incorporating more information into $o_t$, such as the natural language description and HTML representation of a action. We detail our approach in the next section.

### 3.2 WorkflowAgent: Specializing Web Agents Through Fine-Tuning

#### 3.2.1 Collecting Production-Scale Data

We collected a large set of real-world, user-annotated data through an AI workflow documentation software that streamlines the creation of step-by-step guides for web-based tasks (we will reveal the software name in the published paper). This software allows users to record their interactions with the web through a browser extension and converts the interactions into well-annotated instructions. We highlight the following properties of the dataset.

**Domain Coverage.** We consider 20 application domains, including 9 productivity tools, e.g., Office, Calendar, and Gmail; 5 customer relationship management tools, e.g., Salesforce, HubSpot, and Netsuite; 2 financial tools (Intuit, Workday); 2 social platforms (Facebook, LinkedIn), a shopping site (Shopify), and a design tool (Canva). The dataset includes ∼10,000 websites and subdomains.

**Workflows and Steps.** There are ∼20,000 workflows with an average of 11 steps per workflow, totaling ∼200,000 steps. After tokenization, the dataset contains ∼6 billion tokens.

**Data Format.** Each workflow features a high-level user objective and a step-by-step documentation of the action sequence to achieve the task. The objective spans a wide range of topics, such as "add a user in a Salesforce"]" or "invite someone to manage Facebook ad accounts". Each step contains: the current web page's URL, raw HTML-DOM, a natural language description of the action performed, the type of action, and the autogenerated CSS selector to identify the action target.

There are three types of actions in the dataset: *mouse_click_action* for clicking an element; *keyboard_sequence_action* for typing a sequence of characters in an element; and *keyboard_combination_action* for pressing a set of keys together like ctrl+c. There is no scroll action in our action space since all elements are already fully accessible in the captured data. This is because we capture the full DOM from a system perspective, which inherently includes the entire webpage as observed from the backend. To ensure data quality, we remove workflows with invalid selectors and non-English workflows. Example raw data can be found in the supplementary material.

#### 3.2.2 Preprocessing

For WorkflowAgent, we consider an observation space consisting mainly of the URL and HTML-DOM. Specifically, HTML-DOM provides agents with all structural and content information about the web page that are essential for generating the next step and long-term planning. For instance, while a drop-down menu may not be visible on the website before expansion, the agent can detect the menu items from the DOM and determine whether to click and expand it. We do not use accessibility tree because our dataset records only HTML-DOM and CSS selectors corresponding to target elements. Converting HTML to accessibility tree offline requires complex techniques and LLM assistance, which may introduce inaccuracies and noise. Besides, accessibility tree does not generalize across different browsers and devices, and it removes useful DOM-specific attributes such as `class` and `href`, potentially leading to information loss.

Given our observation space, a subsequent problem is that the DOM can be quite long and exceed the context window of prevailing open-source LLMs. To reduce the DOM sizes, we propose a pruning pipeline that maintains the essential structure and content while eliminating redundant or disruptive elements that

could hinder the LLM's understanding. Specifically, we first use the BeautifulSoup library (Richardson, 2007) to remove non-essential components such as metadata, CSS, and JavaScript. Then, we utilize a tag-attribute white list to retain useful tag level information like retaining interactive elements. Since some attribute values can contain random character sequences that do not provide useful information, we propose a novel detection method that removes the attributes with character-to-token-ratio smaller than 2, i.e., $\frac{len(s)}{len(tokenizer(s))} < 2$, where $s$ denotes the value string. Intuitively, if each character in a string is encoded using a separate token, it is highly likely that the string is not semantically meaningful. After pruning, we assign each tag in the HTML with a unique ID by traversing the HTML tree from bottom to top. More details about preprocessing and analysis on the tokenizer-pruning method can be found in Appendix A.1. We also provide the code in the supplementary material. After preprocessing the HTML, we rewrite each step into five lines as follows:

```
1.
Description: Click the "Menu" button to browse all food options
Action: mouse_click_action
Node: 832
Target: <svg class="open-hamburger-icon" node="832" role="img">
```

The first line represents the current time step. The second line is the natural language description of the action, which can help LLMs to learn the rationale behind applying a specific action. The third line is one of the three operations in the action space. The fourth line is the unique ID assigned to the target element. The last line details the HTML tag and attributes, which can be directly obtained from the processed DOM.

Then, our dataset consists of input-output pairs for training next-step prediction. For each pair of data, the input prompt includes the task objective, URL, HTML-DOM, and all past actions in the aforementioned five-line format. The output is the next action represented in the five-line format. We provide an example workflow in Appendix A.2.

### 3.2.3 Fine-Tuning with LoRA

We divide the processed data into two splits. The test split comprises of 1200 workflows with diverse objectives and domains. We use the remaining workflows as the training data to adapt LLMs via standard supervised fine-tuning. Note that we train the agent to predict a single next-step rather than all remaining steps because the environment would change after performing an action. Thus, every action needs an inference call. Moreover, the agent is trained to generate all information in the five-line format described above, including the natural language description to facilitate reasoning.

To reduce fine-tuning cost, we opt for the parameter-efficient method LoRA (Hu et al., 2022) instead of full fine-tuning, since we have not observed significant performance gain by updating more parameters. We also follow previous work (Zhao et al., 2023; Shen et al., 2023) to fine-tune the layernorms in addition to the LoRA adapters. Through hyperparameter tuning, we set the training epoch to 2, effective batch size to 32, LoRA rank to 64, and $\alpha$ to 128. We use a cosine scheduler with 30 warmup steps and a learning rate of 1e-4.

### 3.2.4 Exploring the Design Space

There are multiple design choices for WorkflowAgent that might affect the prediction accuracy, fine-tuning cost, and inference latency. We focus on three aspects and perform detailed ablation studies to find out the optimal modeling and training configurations.

**Pretrained LLM Selection.** Intuitively, the quality of a fine-tuned web agent should be relevant to the quality of the pretained LLM. We identify two axes that are crucial to performance—model architecture and model size—and explore seven open-source LLMs spanning these axes: Llama 3.1 8B (Dubey et al., 2024), Mistral 7B (MistralAI, 2023), Mixtral 8x7B (MistralAI, 2024b), Qwen2 7B (Yang et al., 2024a), Qwen2 57B (Yang et al., 2024a), Qwen2.5 14B (Team, 2024), Qwen2.5 32B (Team, 2024), and Codestral 22B (MistralAI, 2024a). We fine-tune these models with 1 billion training tokens and evaluate their performance on the test split of the dataset we collected.

Given that many of the evaluated LLMs have a maximum context window of approximately 32K, and the processed DOM can exceed this limit, we divide the DOM into chunks that fit into the context window. For

Table 1: Test results of different LLMs fine-tuned on 1B tokens. We highlight the best results for small/medium/large models.

| Model | # Params | EM Before Fine-Tune (%, ↑) | EM After Fine-Tune (%, ↑) |
|---|---|---|---|
| Mistral-7B-Instruct-v0.3 | 7B | 5.13 | 26.31 |
| Qwen2-7B-Instruct | 7B | **7.92** | **38.72** |
| Llama-3.1-Instruct-8B | 8B | 1.88 | 37.42 |
| Qwen2.5-14B-Instruct | 14B | **11.6** | **41.89** |
| Codestral-22B-v0.1 | 22B | 6.08 | 41.25 |
| Qwen2.5-32B-Instruct | 32B | **13.21** | **43.51** |
| Mixtral-8x7B-Instruct-v0.1 | 56B-A12B | 9.82 | 37.49 |
| Qwen2-57B-A14-Instruct | 57B-A14B | 7.51 | 40.10 |

Table 2: Effect of context window.

| Model | Context | EM (%, ↑) |
|---|---|---|
| Qwen2 7B | 32K | 38.72 |
| Qwen2 7B | 65K | 36.22 |
| Qwen2.5 14B | 32K | 41.89 |
| Qwen2.5 14B | 65K | 39.15 |
| Qwen2.5 32B | 32K | 43.51 |
| Qwen2.5 32B | 65K | 41.69 |

fine-tuning, we use the chunk containing the correct target. We only evaluate on tasks whose DOM can fit into the context length. We compute the exact match (EM) metric, which measures the model's accuracy when the selected HTML tag matches the ground truth.

We report the performance of different LLMs before and after fine-tuning in Table 1. Notably, for all models, specialized fine-tuning drastically increases the prediction accuracy. Both before and after fine-tuning, the Qwen family demonstrates better EM across small, medium, and large models. We observe performance gains as model size increases, e.g., Qwen2.5 32B and Mixtral 8x7B outperforms Qwen2.5 14B and Mistral 7B, respectively. However, fine-tuning larger models is significantly more resource-intensive—while Qwen2 7B can be fine-tuned using 8 H100 GPUs in just one day, Qwen2 57B takes over a week using the same hardware configuration. Larger models also incur longer inference times and require multiple GPUs for serving.

**Context Window Length.** We evaluate the models with 65K context window to add additional context (Table 2). On both Qwen2 and Qwen2.5, scaling up the context window leads to approximately 2.5% performance drop, as it becomes harder to pick the correct target given twice as many options to choose from. Using 65K context window also increases the inference time by approximately four times in practice.

**Dataset Size.** Lastly, we study the effect of fine-tuning dataset size on the agent's performance. We sample our training set without replacement into smaller subsets and fine-tune Qwen2-7B-Instruct on them. Results are shown in Table 3. Plotting on a log-linear scale, we observe that there is a roughly 2% performance boost when we double our dataset size.

Table 3: Effect of dataset size.

| # Tokens | EM (%, ↑) |
|---|---|
| 1B | 38.72 |
| 3B | 43.06 |
| 6B | 46.42 |

To sum up, we study the effect of LLM backbone, context window, and dataset size on the agent performance using our collected dataset. We find that scaling parameter count and dataset size improves prediction quality, but the latency and training time of large LLMs can be prohibitive. Based on these insights, we develop two versions of WorkflowAgent using the full 6B-token dataset and a 32K context window: WorkflowAgent-Small is fine-tuned from Qwen2 7B and offers an optimal balance between prediction accuracy, training and inference costs; WorkflowAgent-Large is fine-tuned from Qwen2.5 32B, enabling stronger performance when we have sufficient compute.

# 4 Results

We evaluate WorkflowAgent on three datasets. We first consider the static next-step prediction setting, where performance is evaluated only on the correctness of next step. We show that WorkflowAgent outperforms various general-purpose LLMs on both our real-world dataset and the public benchmark Mind2Web (Deng et al., 2023). Then, we move to the end-to-end task completion setting and develop a multi-agent system based on WorkflowAgent to interact with the environment, which achieves top performance on WebArena (Zhou et al., 2024). Note that we do not perform any task-specific adaptation for Mind2Web and WebArena, even when additional training data is available. This allows us to evaluate the generalization ability of WorkflowAgent.

## 4.1 Self-Collected, Real-World Data

To study whether specialized fine-tuning is indeed beneficial, we first compare the performance of WorkflowAgent with general-purpose baselines on our collected test data. We consider the non-fine-tuned Qwen2 7B, GPT-4o, and GPT-4o mini. We use in-context demonstrations to prompt them to generate actions in the same five-line format as defined in Section 3.2.2. All OpenAI baselines follow the prompt in Appendix A.3.2.

Table 4: WorkflowAgent v.s. non-fine-tuned, general-purpose baselines on the full test set.

| Model | EM (%, ↑) |
|---|---|
| Qwen2 7B | 8.20 |
| GPT-4o mini | 13.26 |
| GPT-4o | 16.02 |
| WorkflowAgent-Small | 46.42 |
| WorkflowAgent-Large | **49.67** |

Table 5: WorkflowAgent vs. OpenAI baselines on test subset with 500 workflows.

| Models | EM (%, ↑) |
|---|---|
| o1-mini | 18.32 |
| o1-preview | 23.79 |
| GPT-4o mini | 14.53 |
| GPT-4o | 17.96 |
| WorkflowAgent-Small | 50.11 |
| WorkflowAgent-Large | **52.60** |

Results on the full 1200 test workflows are shown in Table 4. First, we note that WorkflowAgent significantly outperforms the proprietary GPT-4o and 4o mini. This shows the benefit of specialized fine-tuning over using general-purpose LLMs. Moreover, while the non-fine-tuned Qwen2 performs extremely poorly, fine-tuning with our dataset (WorkflowAgent-Small) boosts its performance by nearly 6×, which highlights the importance of domain-specific data. We also show WorkflowAgent consistently outperforms the general-purpose baselines across commonly seen domains, such as CRM tools, E-commerce platforms, productivity tools, and social platforms in Appendix Figure 2.

As we were wrapping up this work, OpenAI released o1 (OpenAI, 2024b), a series of specialized models for solving complex tasks in science, coding, and math. Since it has better planning ability, we also include it in our baselines. However, we did not run the o1 models on the full test set due to cost and API call limitations. Instead, we subsample 500 workflows and compare with WorkflowAgent. As shown in Table 5, o1-preview performs the best among all general-purpose baselines. However, WorkflowAgent still outperforms it by a wide margin, highlighting the importance of fine-tuning on real-world web navigation data.

It is important to note that WorkflowAgent-Small only has 7B parameters, while WorkflowAgent-Large has 32B parameters, and neither model requires additional scaling during inference. In contrast, most proprietary baselines are typically larger in size and require more compute at inference. This makes WorkflowAgent a better choice in terms of accuracy, latency, and cost.

## 4.2 Mind2Web

Mind2Web (Deng et al., 2023) is a static, text-based dataset for assessing the navigation ability of web agents across different tasks, websites, and domains. Each task features a human demonstration of a real-world workflow, such as booking a hotel on Airbnb. At each step, the agent is asked to predict an operation and a target element as the next action. Performance is measured by element accuracy, which checks if the correct target is selected; action F1 score, which measures operation correctness like text input; step success rate, which evaluates whether both the target element and the operation are correct; and task success rate, indicating all steps are correct.

The original Mind2Web benchmark reports two sets of baselines: (1) multi-stage, multi-choice question-answering agents (i.e., the MindAct family), which use a pretrained element-ranking model to filter out 50 candidate elements from the full DOM and use a separate LLM to recursively select an action from five candidates until one action is chosen; (2) single-stage, generation-based agents (i.e., fine-tuned Flan-T5), which directly generates the operation and the target based on the full DOM. The multi-stage baselines generally show higher metrics than direct generation models, as the element selection process effectively filters out noise, simplifying the task.

Beyond the original baselines, we also consider recent published work such as AWM (Wang et al., 2024b), Synapse (Zheng et al., 2024), and fine-tuned HTML-T5 (Gur et al., 2023). For both single-stage and multi-stage settings, we further categorize the baselines into those leveraging the Mind2Web training data for fine-tuning or in-context demonstrations vs. zero-shot methods. We do not fine-tune WorkflowAgent to show its out-of-distribution generalization ability.

We evaluate WorkflowAgent on both multi-stage QA and direct generation. For the multi-stage setting, we first use the pretrained Mind2Web ranker to obtain the element ranking. Then, given our agent's output, we traverse the sorted list of HTML elements from top to bottom and stop when the agent's generated HTML

Table 6: WorkflowAgent achieves SOTA zero-shot performance on Mind2Web. EA is short for element accuracy, $AF_1$ is short for action $F_1$, SR is short for success rate. Numbers are bolded for each method category.

| Method | | Cross-Task | | | | Cross-Website | | | | Cross-Domain | | | |
|---|---|---|---|---|---|---|---|---|---|---|---|---|---|
| | | EA | $AF_1$ | Step SR | Task SR | EA | $AF_1$ | Step SR | Task SR | EA | $AF_1$ | Step SR | Task SR |
| | **Uses M2W Train Set** | | | | | | | | | | | | |
| | MindAct (Flan-T5$_B$) | 43.6 | 76.8 | 41.0 | 4.0 | 32.1 | 67.6 | 29.5 | 1.7 | 33.9 | 67.3 | 31.6 | 1.6 |
| | MindAct (Flan-T5$_L$) | 53.4 | 75.7 | 50.3 | 7.1 | 39.2 | 67.1 | 35.3 | 1.1 | 39.7 | 67.2 | 37.3 | 2.7 |
| | MindAct (Flan-T5$_{XL}$) | 55.1 | 75.7 | 52.0 | 5.2 | 42.0 | 65.2 | 38.9 | 5.1 | 42.1 | 66.5 | 39.6 | 2.9 |
| | AWM-offline (GPT-4) | 50.6 | 57.3 | 45.1 | 4.8 | 41.4 | 46.2 | 33.7 | 2.3 | 36.4 | 41.6 | 32.6 | 0.7 |
| Multi-Stage QA | HTML-T5-XL | **60.6** | **81.7** | **57.8** | **10.3** | **47.6** | **71.9** | **42.9** | **5.6** | **50.2** | **74.9** | **48.3** | **5.1** |
| | **Zero-Shot** | | | | | | | | | | | | |
| | MindAct (GPT-4) | 41.6 | **60.6** | 36.2 | 2.0 | 35.8 | 51.1 | 30.1 | 2.0 | 21.6 | 52.8 | 18.6 | 1.0 |
| | AWM-online (GPT-4) | 50.0 | 56.4 | 43.6 | **4.0** | 42.1 | 45.1 | 33.9 | 1.6 | 40.9 | 46.3 | 35.5 | **1.7** |
| | WorkflowAgent Small (**Ours**) | 42.6 | 50.1 | 39.7 | 0 | 44.9 | 50.1 | 41.6 | 0.6 | 44.1 | 51.4 | 41.4 | 0 |
| | WorkflowAgent Large (**Ours**) | **53.5** | 52.9 | **51.2** | 0 | **53.4** | 52.8 | 51.3 | 2.3 | 53.3 | 54.7 | 51.2 | 0 |
| | **Uses M2W Train Set** | | | | | | | | | | | | |
| Direct Genera-tion | Flan-T5$_B$ | 20.2 | **52.0** | 17.5 | 0 | 13.9 | **44.7** | 11.0 | 0 | 14.2 | **44.7** | 11.9 | 0.4 |
| | Synapse (GPT-3.5) | **34.0** | - | **30.6** | **2.4** | **29.1** | - | **24.2** | **0.6** | **29.6** | - | **26.4** | **1.5** |
| | **Zero-Shot** | | | | | | | | | | | | |
| | WorkflowAgent Small (**Ours**) | 28.6 | 50.1 | 26.8 | 0 | 27.6 | 50.1 | 25.6 | 0 | 32.0 | 51.4 | 29.9 | 0 |
| | WorkflowAgent Large (**Ours**) | **38.0** | **52.9** | **35.6** | 0 | **34.1** | **52.7** | **32.5** | 0 | **39.4** | **54.7** | **37.3** | 0 |

element is a child node of the element. We then replace WorkflowAgent's prediction by the element. For direct generation, we simply compare the output of our agent with the ground-truth operation and target.

Results are shown in Table 6. For the multi-stage setting, WorkflowAgent-Large achieves the best overall zero-shot performance. Our element accuracies and step success rates are also competitive with the best fine-tuned baseline, HTML-T5-XL, on cross-website and cross-domain tasks. However, our task success rates are not satisfactory, which is due to the distribution differences between our training data and the Mind2Web data. Upon inspection, we find that the primary failure cases of our models are (1) predicting the children element of the ground truth instead of the ground truth; (2) predicting another element with identical function but is different from the ground truth; and (3) our agent tends to decompose type actions into click followed by type actions. In many cases, we actually correctly predict the action description. These situations can be addressed by improving the evaluation procedure, which we discuss in Appendix A.4.2. By accounting for the children element of the ground truth and comparing also the tag and text attributes, the task success rates and element accuracies for our agent increase by ∼8%.

As for direct generation, WorkflowAgent-Large outperforms all existing baselines. Our step success rates are 2-3× higher than fine-tuned Flan-T5 and show an improvement of 5-10% over Synapse, which utilizes GPT-3.5. We attribute WorkflowAgent's strong performance to the diversity and high quality of the workflows in our dataset. Relatedly, the three test sets (Cross-Task, Cross-Website, Cross-Domain) are designed to capture different degrees of domain generalization difficulty. Since we do not train on Mind2Web data, the performance is similar across all three test sets.

While the Mind2Web results are promising, we note that a limitation of static, text-based benchmark is that the ground-truth evaluation does not account for different action sequences that could reach the same goal. For instance, to book a flight, one can first enter the destination or first choose the departure date, but the ground truth trajectory only accounts for one possibility. Considering this, we also evaluated WorkflowAgent on a dynamic benchmark WebArena (Zhou et al., 2024).

### 4.3 End-to-End Task Execution on WebArena

WebArena (Zhou et al., 2024) features 812 web navigation tasks across five domains: E-commerce (OneStop-Shop), social forums (Reddit), software development (GitLab), content management (CMS), and online map (OpenStreetMap). Unlike the static Mind2Web, it implements a dynamic environment for agents to interact with. However, WebArena only accepts actions specified in the *accessibility tree* format, which WorkflowA-gent does not natively handle, and it includes information-retrieval tasks, which our agent is not explicitly trained for. To address these challenges, we design a multi-agent system leverages both WorkflowAgent and

Table 7: Task success rates (SR) on WebArena and score breakdown on five web domains. WorkflowAgent consistently outperforms existing text baselines, often improving the previous-best results by more than 5%.

| Method | LLM | Total SR | Shopping | CMS | Reddit | GitLab | Maps |
|---|---|---|---|---|---|---|---|
| AutoWebGLM | ChatGLM3 6B | 18.2 | - | - | - | - | - |
| AutoEval | GPT-4 | 20.2 | 25.5 | 18.1 | 25.4 | 28.6 | 31.9 |
| BrowserGym$_{axtree}$ | GPT-4 | 15.0 | 17.2 | 14.8 | 20.2 | 19.0 | 25.5 |
| SteP | GPT-4 | 33.0 | 37.0 | 24.0 | 59.0 | 32.0 | 30.0 |
| AWM | GPT-4 | 35.5 | 30.8 | 29.1 | 50.9 | 31.8 | 43.3 |
| Tree Search | GPT-4o | 19.2 | - | - | - | - | - |
| WebPilot | GPT-4o | 37.2 | 36.9 | 24.7 | 65.1 | 39.4 | 33.9 |
| Broswing+API Hybrid Agent | GPT-4o | 35.8 | 25.7 | 41.2 | 28.3 | 44.4 | 45.9 |
| AgentOccam with Judge | GPT-4-Turbo | 45.7 | 43.3 | **46.2** | 67.0 | 38.9 | 52.3 |
| Multi-Agent System (**Ours**) | WorkflowAgent-Small + GPT-4o | 51.3 | **48.1** | 35.5 | 70.2 | 58.8 | 51.9 |
| | WorkflowAgent-Large + GPT-4o | **53.0** | 45.8 | 37.9 | **73.7** | **59.7** | **56.3** |

Table 8: We replace WorkflowAgent with GPT-4o in our four-stage pipeline to study how much WorkflowAgent contributes to the performance. The success rates drop significantly for all domains.

| Method | LLM | Total SR | Shopping | CMS | Reddit | GitLab | Maps |
|---|---|---|---|---|---|---|---|
| Single-Agent | GPT-4o | 34.2 | 31.9 | 21.3 | 44.7 | 38.2 | 42.6 |
| Multi-Agent | WorkflowAgent-Small + GPT-4o | **51.3** | **48.1** | **35.5** | **70.2** | **58.8** | **51.9** |

GPT-4o to bridge the representation gap and tackle information-retrieval. However, we show in the ablation study (Figure 8) that the performance gain of this multi-agent system mainly comes from WorkflowAgent.

The system contains four stages: (1) planning: we use general-purpose GPT-4o to produce a task plan based on the user objective; (2) action generation: WorkflowAgent outputs an action given the observation; (3) action execution: GPT-4o maps the agent's output in HTML to the accessibility tree format, which is then executed by the environment; (4) termination check: GPT-4o compares the current state with the task plan to decide whether the task is completed or not. We consider top-performing, text-only agents on the WebArena leaderboard as baselines. OpenAI Operator (OpenAI, 2025) and AWA 1.5 (JaceAI, 2024) are excluded because they are multi-modal, and the former came out after we finish our work. More evaluation details can be found in Appendix A.5.

The results are shown in Table 7. Compared with existing text-only baselines, our multi-agent system obtains the highest task success rate in 4 of 5 categories, leading to 7.3% performance improvements in total success rate over the previous-best GPT-4-Turbo-based AgentOccam (Yang et al., 2024b). In particular, on Reddit and GitLab tasks where the domains are more realistic and thus closer to the ones in our training data, WorkflowAgent demonstrates stronger generalization ability and higher task success rates than in other domains. Despite known issues with combobox selection and the absence of scroll actions in our training data, our agent effectively navigates these challenges through strategic keyboard actions (see how in Appendix A.5.2).

To better understand the contribution of WorkflowAgent to the multi-agent system, we perform an ablation study that leverages GPT-4o for all four stages of the proposed pipeline. Given the large number of tasks in WebArena and the evaluation costs associated with using a multi-agent system, we perform our ablation studies using WorkflowAgent-Small. As shown in Table 8, using WorkflowAgent consistently outperforms the GPT-4o-only setting. This shows that our strong performance on WebArena can be mainly attributed to the action generation process of WorkflowAgent.

# 5 Limitations and Future Work

While our work demonstrates strong empirical performance on benchmark datasets and provides valuable insights into fine-tuning web agents, it has several limitations that open up opportunities for future research.

First, while our approach uses HTML-DOM for web representation, alternative formats such as accessibility tree may offer advantages in certain scenarios, particularly in reducing context length. Exploring the impact of different web representations remains an important direction for future work. Besides, the long-context nature of DOMs presents great challenges in adapting LLMs. As a next step, we aim to enable WorkflowAgent to compare and reason over multiple DOM chunks. This requires integrating a memory and a decision-making component. We also plan to explore retrieval-augmented generation (RAG) techniques to improve information retrieval across chunks. Lastly, we aim to expand WorkflowAgent's capabilities to handle multi-modal inputs. This would significantly broaden its applicability across different visual contexts, making it more versatile and robust in real-world web environments.

## 6 Conclusion

In this work, we explore how fine-tuning open-source LLMs with high-quality real-world workflow data can benefit developing specialized web agents. We present WorkflowAgent, which consistently outperforms existing methods that prompt proprietary models in various evaluation settings and benchmarks. We also provide empirical insights into data processing and model fine-tuning.

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

## A    Appendix

### A.1    Preprocessing

#### A.1.1    Pruning Pipeline

The code for preprocessing, fine-tuning, and inference can be found in the supplementary material and will be released on GitHub.

#### A.1.2    Tokenizer Pruning

In this section, we provide more details on the tokenizer-based detection method to remove random character strings. The rationale behind our approach is based on the observation that typical English words consist of more than two characters. Assuming the token count is $t$ and the character count is $s$, this means that when $t = 1$, $s \geq 2$, leading to $\frac{s}{t} \geq 2$. By setting the pruning threshold to 2 and removing tag attributes with $\frac{s}{t} < 2$, we aim to eliminate strings composed solely of single-character tokens, which are likely to be nonsensical.

In our actual implementation, we employ this technique only for tag attributes with $s > 32$, being more lenient for shorter attributes. To show that this tokenizer pruning strategy is effective and to study the performance across different tokenizers and pruning thresholds, we perform the following experiments.

We take three tokenizers from different models: Qwen2-7B-Instruct, Mistral-7B-Instruct-v0.3, and Meta-Llama-3-8B. For each tokenizer, we vary the pruning thresholds across a set of values: $\{1.5, 1.75, 2, 2.25, 2.5\}$. Note that it is meaningless to study overly small thresholds (e.g., it is impossible to have $\frac{s}{t} < 1$) or overly large thresholds (e.g., $\frac{s}{t} < 3$ could result in the loss of meaningful attributes, as many English words contain three letters). We randomly sample 1000 DOMs from our proprietary test dataset, apply our standard pruning pipeline followed by tokenizer pruning, and then perform three analysis:

- False positives: we use the Python `enchant` library to detect if there are meanful English words within the pruned strings. Note that even though these are actual words, many of them are related to DOM structure and can be safely ignored. Still, we count them as false positives since the tokenizer method is designed to remove random character strings.

- Average $s$ and $t$ for the entire DOM before and after tokenizer pruning: this is for understanding the reduction in content length.

- Lastly, we sort tags and attributes by the frequency of being pruned to identify patterns.

As shown in Table 9, there is a clear trade-off between precision and context reduction: greater reductions in content length tend to result in higher false positive rates. While different tokenizers exhibit varying sensitivities to the pruning thresholds, a threshold of 2 achieves the most balanced trade-off, which aligns with our intuition. We then list the top-5 tag-attribute pairs most frequently pruned under threshold 2 along with their pruning counts:

- Qwen: ('div', 'class'): 3188, ('span', 'class'): 11426, ('a', 'href'): 8802, ('button', 'class'): 6844, ('i', 'class'): 5010

- Mistral: ('div', 'class'): 5288, ('span', 'class'): 15824, ('a', 'href'): 12948, ('button', 'class'): 7998, ('svg', 'class'): 5871

- Llama: ('div', 'class'): 29559, ('span', 'class'): 8823,('button', 'class'): 5889, ('i', 'class'): 4608, ('svg', 'class'): 2577

Attributes such as 'class' often contain random character strings and are frequently pruned. However, we observe differences in how tokenizers handle the href attribute: both Qwen and Mistral tokenizers tend to prune it away, whereas the Llama tokenizer preserves it, indicating its better capability in tokenizing URLs.

Table 9: Tokenizer pruning analysis.

| Tokenizer | Prune Threshold | False Positive (%) ↓ | Before Pruning (K) | | After Pruning (K) | | |
|---|---|---|---|---|---|---|---|
| | | | $s$ | $t$ | $s$ | $t$ | $\Delta t$ |
| Qwen2-7B-Instruct | 1.5 | 0.025 | | | 221.4 | 77.11 | 2.03 |
| | 1.75 | 0.013 | | | 217.3 | 74.67 | 4.47 |
| | 2 | 0.18 | 224.3 | 79.14 | 215.7 | 73.89 | 5.21 |
| | 2.25 | 0.36 | | | 213.9 | 73.13 | 6.01 |
| | 2.5 | 0.38 | | | 210.0 | 71.63 | 7.51 |
| Mistral-7B-Instruct-v0.3 | 1.5 | 0.012 | | | 219.5 | 87.10 | 3.44 |
| | 1.75 | 0.18 | | | 216.1 | 85.07 | 5.47 |
| | 2 | 0.44 | 224.3 | 90.54 | 212.7 | 83.40 | 7.14 |
| | 2.25 | 0.49 | | | 205.3 | 80.20 | 10.34 |
| | 2.5 | 11.28 | | | 190.3 | 74.44 | 16.10 |
| Meta-Llama-3-8B | 1.5 | 0.0097 | | | 223.1 | 70.60 | 0.84 |
| | 1.75 | 0.012 | | | 218.3 | 67.85 | 3.59 |
| | 2 | 0.035 | 224.3 | 71.44 | 216.8 | 67.09 | 3.43 |
| | 2.25 | 0.043 | | | 215.2 | 66.41 | 5.03 |
| | 2.5 | 0.10 | | | 212.7 | 65.46 | 5.98 |

Although we currently use the Qwen tokenizer in our preprocessing pipeline to align with the backbone model of WorkflowAgent, the Llama tokenizer can be a compelling alternative for future consideration since it is better at recognizing URLs and producing shorter token sequences. In general, we believe developing specialized models can be important to achieve strong results, as evidenced in prior works (Shen et al., 2024a; Tu et al., 2022; Shen et al., 2022; Roberts et al., 2021).

## A.2 Example Prompt and Label for WorkflowAgent

```
Objective: Grant delegation access to another user in Gmail settings.
URL: https://mail.google.com/mail/u/0/
Observation: {processed dom}
Step-by-step guide:
1.
Description: Click "See all settings"
Action: mouse_click_action
Node: 254
Target: <button class="Tj" node="254">
2.
Description: Click "Accounts"
Action: mouse_click_action
Node: 2625
Target: <a class="f0 LJOhwe" href="https://mail.google.com/mail/u/0/?
tab=#settings/accounts" node="2625" role="tab">
3.
Description: Click "Add another account"
Action: mouse_click_action
Node: 1215
Target: 
```

### A.3   OpenAI Prompts

#### A.3.1   Data Preparation

Below shows the prompt to generate step descriptions.

```
You are navigating a webpage to achieve an objective. Given the
objective, a list of the previous actions, the current action, and a
screenshot of the current action on the webpage. The objective and
previous steps are only here to ground the current step, the current
action and its screenshot are the most useful to your task. Give me
a concise description of the current action being done on the webpage.
You should look at the part of the webpage with the red circle, this is
where the user clicked for the current action. Describe this action
and ensure your response is in the same format, concise, coherent.
Use any relevant information in the image to ground the action
description. Your response should NOT use any json or markdown formatting.
The response should be a single sentence that starts with an action verb.
For example, 'Click on the 'SUBMIT' button.'
```

**Regenerated Action Descriptions.** We provide a few examples of generated action descriptions using GPT-4o.

- "Click on the Submit button."
- "Type in the name of the item."
- "Double-click on the highlighted text."

#### A.3.2   Proprietary Benchmark Baselines

Below shows the prompt for all OpenAI baselines. The text is the prepend for every input to which we append the task input with the corresponding objective, URL, DOM, and action history.

```
You are an autonomous intelligent agent tasked with solving web-based
tasks. These tasks will be accomplished through the use of
specific actions you can issue.
Here's the information you'll have:
- The user's objective: This is the task you're trying to complete.
- The current web page's URL: This is the page you're currently navigating.
- Part of the current web page's HTML: Each element is assigned in
descending order with an unique ID, denoted by the attribute \"node\".
The actions you can perform include:
- mouse_click_action: click
- keyboard_sequence_action: type a sequence of characters
- keyboard_combination_action: press a set of keys together
(e.g., hotkey like ctrl+c)
You will generate a step-by-step guide to complete the task based on the
given information. You will only produce a SINGLE next step.
Do NOT use additional punctuation, or any markdown formatting.
The output should be in the following format:
Description: Click \"Users\"
Action: mouse_click_action
Node: 93
Target: <a node=\"93\" class=\"slds-tree__item-label\">
Now complete the following task by generating the next step.
{task input}
```

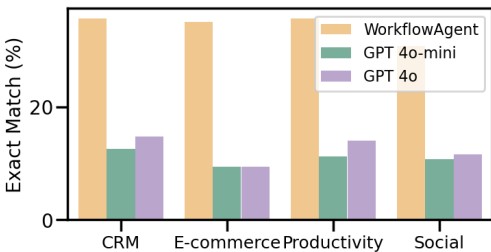

Figure 2: Exact Match (EM) comparison between WorkflowAgent-Small and OpenAI models across different types of websites.

### A.4  Mind2Web Experiment Details

#### A.4.1  Preprocessing

**Data and Label Conversion.** To apply WorkflowAgent to Mind2Web data, we first re-process the provided DOM using the procedure detailed in Section 3.2.2. We store a map between our node ID and the backend ID given in the dataset. Then, we transform the history action provided in the dataset to our 5-line format. After WorkflowAgent generates the next step, we check the backend ID of the provided label and map it to the node ID in our processed DOM. We then compare this label with the target node ID generated by WorkflowAgent. We provide the code for the DOM processing and label conversion process in the supplementary material and will release them later.

**DOM Chunking and Action Generation.** When the DOM length exceeds the 32K context window, we chunk the DOM sequentially and run the prediction workflow on each piece. For each piece of DOM, we call WorkflowAgent five times to obtain five valid actions. We then aggregate all possible actions and select the one with the highest number of appearances. We use the following generation configuration: do_sample=True, top_p=0.95, temperature=0.6.

#### A.4.2  Refined Evaluation

As mentioned in the main text, we improve the Mind2Web evaluation from two perspectives:

- Subchild label relaxation: We hypothesize that the distribution gap between our training data for WorkflowAgent and the Mind2Web test set could be due to Mind2Web preferring ancestor/parent nodes in the HTML tree, while WorkflowAgent's training data prefers lower HTML elements. To this effect, we relax the Mind2Web set of positive candidates to include not only the positive candidates, but also their children (direct children and grandchildren).

- Attribute matching: Direct generation setting enables higher degree of freedom in element selection. To address scenarios where the predicted element has the same function as the ground truth but is in a different location, we enhance the direct generation evaluation by introducing an element attribute comparison step. Rather than merely comparing the node ID of the predicted and the ground truth elements, we also evaluate the tag and text attributes (e.g., the text displayed on a button). If these attributes match, we consider the prediction to be correct as it has identical functionality.

Lastly, we note that in Mind2Web, whenever there is a `textarea` or an `input` tag, the expected behavior is to directly execute the type action. However, our model is trained to first click on the input element and then perform the type action. Thus, for actions predicted on `textarea` or `input` tags, we adjust our model to replace click actions with type actions and then compare with the ground truths.

Table 10 presents the improved performance of WorkflowAgent after refining the evaluation method, showing significant gains in both settings. We find that the label relaxation strategy helps bridge part of the distribution gap, and our multi-stage pipeline effectively covers most of the gains from this label relaxation strategy by using the Mind2Web ranker. However, inspecting cases that are not covered by label relaxation,

Table 10: We also refine the evaluation procedure to better reflect WorkflowAgent's capacity.

| Models | | Eval | Cross-Task | | | | Cross-Website | | | | Cross-Domain | | | |
|---|---|---|---|---|---|---|---|---|---|---|---|---|---|---|
| | | | EA | AF$_1$ | Step SR | Task SR | EA | AF$_1$ | Step SR | Task SR | EA | AF$_1$ | Step SR | Task SR |
| **Multi-Stage** | WorkflowAgent-Small | M2W | 42.6 | 50.1 | 39.7 | 0 | 44.9 | 50.1 | 41.6 | 0.6 | 44.1 | 51.4 | 41.4 | 0 |
| | | M2W + Subchild | 42.6 | 50.1 | 39.8 | 0 | 45.2 | 50.1 | 41.5 | 0.6 | 44.3 | 51.4 | 41.6 | 0 |
| | WorkflowAgent-Large | M2W | 53.5 | 52.9 | 51.2 | 0 | 53.4 | 52.8 | 51.3 | 2.3 | 53.3 | 54.7 | 51.2 | 0 |
| | | M2W + Subchild | 53.8 | 52.9 | 51.3 | 0 | 54.0 | 52.8 | 51.9 | 2.3 | 53.5 | 54.7 | 51.4 | 0 |
| **Direct Gen** | WorkflowAgent-Small | M2W | 28.6 | 50.1 | 26.8 | 0 | 27.6 | 50.1 | 25.6 | 0 | 32.0 | 51.4 | 29.9 | 0 |
| | | M2W + Subchild + Attr Match | 48.8 | 60.8 | 48.3 | 5.5 | 58.0 | 66.2 | 56.7 | 6.8 | 52.9 | 62.1 | 52.4 | 6.5 |
| | WorkflowAgent-Large | M2W | 38.0 | 52.9 | 35.6 | 0 | 34.1 | 52.7 | 32.5 | 0 | 39.4 | 54.7 | 37.3 | 0 |
| | | M2W + Subchild + Attr Match | 58.0 | 63.8 | 52.0 | 5.7 | 67.3 | 69.4 | 59.8 | 11.8 | 62.0 | 63.7 | 52.9 | 10.8 |

we found that there still remains a distribution gap. As a result, there is large room for improving the evaluation criteria of text-based benchmark to bridge this gap.

## A.5 WebArena Experiment Details

### A.5.1 Four-Stage Pipeline

For the most up-to-date prompts, please refer to our GitHub.

**Stage 1:** GPT-4o refines the intent. We use the following prompt:

```
I have a simple task objective related to {domain}, rewrite
it into a single paragraph of detailed step-by-step actions to achieve
the task. When revising the objective, follow the rules:\\
- Assume you are already on the correct starting website and are logged in.\\
- Do not include any newlines, tabs, step numbers in the rewritten objective.\\
- Follow the example as much as possible.\\
{In-context demonstrations for domain rules}\\
Here is an example:\\
Simple Task Objective: {in-context demonstration}\\
Detailed Task Objective: {in-context demonstrations}\\
Now, rewrite the following objective:
```

**Stage 2:** We process the environment-generated DOM using our preprocessing procedure. When the DOM length exceeds the 32K context window, we chunk the DOM sequentially and run the prediction workflow on each piece. For each piece of DOM, we call WorkflowAgent multiple times to obtain multiple valid actions. We use the following generation configuration: do_sample=True, top_p=0.95, temperature=0.6. We then aggregate all possible actions, pick the top candidates, and prompt GPT-4o to select the best candidate using the following prompt:

```
You are an autonomous agent helping users to solve web-based tasks.
These tasks will be accomplished through series of actions.
The information you'll have includes:\\
- The user's objective\\
- The current web page's URL\\
- The current web page's accessibility tree\\
- Previous steps performed by the user, where each step includes a
description of the action and the target web element\\
- Several proposed next steps, labeled by ``No."\\
Your goal is to select the best next step that can complete the task
and output this candidate's number, follow the following rules:\\
- Do not repeat previous steps\\
- Reject candidates with incorrect intentions, e.g., searching for an
item different from the one specified in the objective\\
```

```
- Reject candidates with factual errors, e.g., the description and
the chosen web target do not match\\
- Only output a single number after to represent the selected candidate
but not explanation\\
Now analyze the following case:
```

**Stage 3:** GPT-4o maps the agent output to accessibility tree format using the following prompt:

```
You are an autonomous agent helping users to solve web-based tasks.
These tasks will be accomplished through series of actions.
The information you'll have includes:\\
- The user's objective\\
- The current web page's URL\\
- A snippit of the current web page's HTML\\
- A snippit of the current web page's accessibility tree\\
- Previous steps performed by the user\\
Your goal is to translate a proposed next step, which consists of an
action and a HTML element, into the following format:\\
- `click [accessibility tree id]': This action clicks on an interactive
(non-static) element with a specific id. Note this id is the number inside
``[]" in the accessibility tree, not the HTML attribute ``node". Brackets are
required in the response. For example, a valid response is ``click [1234]"\\
- `type [accessibility tree id] [content]': Use this to type the content
into the field with a specific id in the accessibility tree.
For example, a valid response is ``type [1234] [New York]".
The second bracket should include everything that needs to appear
in the textbox, but not only the added content. Do not change the letter case\\
- `press [key_comb]': Simulates pressing a key combination on the keyboard
(e.g., press [PageDown], press [Enter])\\
- `go_back`: Return this when the current web page does not contain useful
information and the user should go back to the previous web page\\
When mapping the next step into actions in the above formats, follow the
following rules:\\
- Take the user's objective into consideration, so the action must help
complete the task\\
- Do not repeat previous steps\\
- Only output a single step in the above format but not explanation\\
Note also: {in-context demonstration of rules}\\
Now analyze the following case:
```

The action is then returned to the environment for execution.

**Stage 4:** GPT-4o evaluates if the task objective is achieved. For operational tasks, if the task is completed, nothing is returned. For information seeking tasks, if the task is completed, GPT-4o retrieves the answer to the question. The prompt looks like the following:

```
You are an autonomous agent helping users to solve web-based tasks.
These tasks will be accomplished through series of actions.
The information you'll have includes:\\
- The user's task, including a high-level objective and a more detailed
illustration\\
- The current web page's URL and accessibility tree\\
- Previous steps performed by the user, where each step includes a
description of the action and the target web element\\
You should follow the rules: {in-context demonstration rules}\\
You will decide whether the task specified by the high-level objective
```

```
is completed (which means the **last** step of the detailed instruction
is completed and the current webpage completes the task) and respond
``completed" or ``incomplete". If the task requires returning a number
or a string and the answer can be obtained in the current webpage,
reply ``completed, [answer]" where ``[answer]" is the number or string.
If the task requires finding a webpage and the current webpage satisfies
the requirement, reply ``completed, [answer]" where ``[answer]" is the
current URL. Now analyze the following case. First provide the reasonings.
Then summarize the answer with ``Summary:", followed by ``completed" or
``incomplete", followed by the answer to the question if applicable.
Do not include newlines after ``Summary:".
```

### A.5.2 Scrolling Actions and Combobox Selection

In our data collection process, we capture the full DOM from a system perspective, which inherently includes the entire webpage as observed from the backend. This method differs from user-centric data collection, where only the elements within the visible browser viewport are captured. Consequently, there is no concept of scrolling in our training datasets since all elements are already fully accessible in the captured data.

However, we recognize the importance of scroll actions in solving WebArena from a user perspective. To address this, before issuing any action to the environment, our multi-agent system includes a viewport check that uses the bounding box position to determine if the target element is within the visible webpage area. If not, the system manually inserts necessary scroll actions to bring the element into view. This ensures accurate interaction with web elements in a typical user scenario.

To handle combox selection, our agent discovers a workaround that bypasses the need for scrolling through comboboxes. Specifically, after clicking on the combobox, it types the name of the desired item in the combobox, which brings the item to the top of the dropdown menu. Then, the agent can simply click the item or press Enter. This approach avoids the need for scrolling and is especially effective in densely populated lists. It improves the task success rate on a large number of Map, Reddit, and GitLab tasks.

### A.5.3 GPT-4o-Only Setting

When we use GPT-4o for stage 2, we use the following prompt:

```
You are an autonomous intelligent agent tasked with solving web-based tasks.
These tasks will be accomplished through the use of specific actions you can
issue. Here's the information you'll have:\\
- The user's objective: This is the task you're trying to complete.\\
- The current web page's URL: This is the page you're currently navigating.\\
- The current web page's HTML: Each element is assigned with an unique ID,
denoted by the attribute ``node".\\
The actions you can perform include:\\
- mouse_click_action: click\\
- keyboard_sequence_action: type a sequence of characters\\
- keyboard_combination_action: press a set of keys together
(e.g., hotkey like ctrl+c)\\
You will generate a step-by-step guide to complete the task based on the given
information. At each step, you can perform only one action to one web element.
The output should be in the correct format: a single step consisting of a text
description, an action, as well as the node and HTML of the target web element
to perform the action. Be coherent, concise, and accurate in your response.
Do NOT use any special characters (e.g., ``*", ``#", etc.) in your response.
Follow EXACTLY the format of the response below.\\
Here is one example:\\
Description: click ``Users"\\
Action: mouse_click_action\\
Node: 93\\
Target: <a class=``slds-tree__item-label" node=``93">\\
Now complete the following task by generating a single next step:
```

### A.5.4 Results

Below, we provide example action trajectories produced by our agent and explanations to shed light on its decision making process.

Domain: Shopping

Task ID: 466

Intent: Add 2 Hawaiian Bamboo Orchid Roots #zc50 - by Discount Hawaiian Gifts to my wish list.

Actions:

1. click [1585] where [1585] is combobox 'Search' autocomplete: both hasPopup: listbox required: False expanded: False → Click on the search bar.

2. type [1585] [Hawaiian Bamboo Orchid Roots #zc50 - by Discount Hawaiian Gifts ] where [1585] is combobox 'Search' focused: True autocomplete: both hasPopup: listbox required: False expanded: True → Type the target item name.

3. press [enter] → Do the search.

4. scroll [down] → Scroll down the page to find out the target item.

5. click [4859] where [4859] is link '2 Hawaiian Bamboo Orchid Roots #zc50 - by Discount Hawaiian Gifts' → Open the item page.

6. click [6733] where [6733] is link 'Add to Wish List'→ Add item to wish list.

Domain: CMS

Task ID: 244

Intent: Show me the email address of the customer who is the most unhappy with Olivia zip jacket.

Actions:

1. click [108] where [108] is link 'MARKETING' → Click on the "MARKETING" button in the side bar.

2. click [419] where [419] is link 'All Reviews → Clicking the side bar shows more option. We can then click on the "All Reviews" button.

3. click [2195] where [2195] is button 'Reset Filter' → Remove all previous set filters.

4. type [4435] [olivia] where [4435] is textbox '' required: False → In the search box, type "olivia".

5. press [enter]→ Do the search

6. stop [emma.lopez@gmail.com] → Inspect all listed reviews and return the email of the negative review.

Domain: Reddit

Task ID: 403

Intent: Change my reddit bio to "Awesome Prompt Artist"

Actions:

1. click [12] where [12] is button 'MarvelsGrantMan136' hasPopup: menu expanded: False → Click my user name on the top right corner.

2. click [249] where [249] is link 'Profile' → After the drop-down menu shows up, click "Profile".

3. click [886] where [886] is link 'Edit biography' → Click the "Edit biography" button.

4. type [2456] [Awesome Prompt Artist] where [2456] is textbox 'Biography' required: False describedby: user_biography_biography_help → Type the target content.

5. click [2474] where [2474] is button 'Save' → Save the new profile.

Domain: GitLab

Task ID: 293

Intent: Show me the command to clone Super_Awesome_Robot with SSH.

Actions:

1. click [1507] where [1507] is textbox 'Search GitLab' required: False → Click on the search bar.

2. type [1516] [Super_Awesome_Robot] where [1516] is searchbox 'Search GitLab' focused: True describedby: search-input-description → Type the repo name in the search bar.

3. click [2082] where [2082] is link 'Convex Eggtart / Super_Awesome_Robot' → Click on the correct repo.

4. click [2699] where [2699] is link 'Clone' → Click on the "Clone" button.

5. stop [git clone ssh://git@metis.lti.cs.cmu:2222/convexegg/super_awesome_robot.git] → Read the command from the pop-up window.

Domain: Maps

Task ID: 7

Intent: Tell me the full address of all international airports that are within a driving distance of 50 km to Carnegie Mellon University.

Actions:

1. click [35] where [35] is textbox 'Search' focused: True required: False → Click on the search box.

2. type [35] [airport Pittsburgh] where [35] is textbox 'Search' focused: True required: False → Type "airport Pittsburgh" in the search box.

3. stop [Pittsburgh International Airport, Airport Boulevard, Findlay Township, Allegheny County, 15231, United States.] → Return "Pittsburgh International Airport, Airport Boulevard, Findlay Township, Allegheny County, 15231, United States." as the answer.

