# OpenReview forum: "WorkflowAgent: Towards Specialized Web Agents Using Production-Scale Workflow Data"
_TMLR — Rejected by TMLR_

### Review · Reviewer_Kw62 · 2025-03-16

**Summary Of Contributions:**

This paper studies the effect of fine-tuning with production-scale workflow data on the performance of web agents. Using an AI workflow documentation software, the authors collected a large set of workflow data representing action sequences executed by real users in real web environments, encompassing a diverse spectrum of websites and a total of 6B tokens. With this amount of data, the authors are able to significantly outperform the prior state-of-the-art models both open-source and closed-source. Additionally, the authors also carried out some useful ablation experiments studying the effect of the amount of data, the choices of base models and sizes of the base models, and the effect of context window etc.

**Audience:**

Yes

**Claims And Evidence:**

Yes

**Requested Changes:**

As mentioned above, the main contribution of this paper to the research community seems to be its engineering efforts in collecting a large-scale data and using it to fine-tune a state-of-the-art model. While I understand that the data might not be shared due to privacy concerns, it would be helpful if the authors can clarify whether the model weights will be available for the open-source community. While I understand the authors do plan to share the code, but it seems that data and model weights are more of the unique contribution here.


I would be curious to see a more careful error analysis on WebArena. I know that WebArena sometimes has errors in the tasks, so would be curious to see how many of the mistakes come from the environment and how many come from the model itself. It might be helpful for the open source community to learn what are the major bottlenecks and error types of the state-of-the-art models.

In general, I think all claims are well supported by empirical evidence. All of the 4 contributions seem to be supported by empirical evidence. However, to make the claim even more scientific (in particular, the claim that "we design an effective HTML preprocessing pipeline that balances between preserving essential information and minimizing context length"), it might be helpful for the paper to compare with prior baselines in refining HTML such as https://arxiv.org/abs/2307.12856 and https://arxiv.org/abs/2404.03648 .

As mentioned above, a potential weakness of the paper is that the impact of the work seems mostly limited to the domain of web agents where the authors collected a large scale dataset for performing fine-tuning, and it is unclear what the takeaway message is for broader research community.

**Strengths And Weaknesses:**

**Stregths:**
I really appreciate the engineering efforts that the authors have taken in terms of carrying out the study, including the collection of a very large scale real-world workflow dataset of 6B tokens.

I think it's meaningful to show that one does not need a super large model (like GPT4-o) to be able to solve regular web tasks like the onces in webarena. With the right domain-specific data, even a small model like Qwen2.5-7B can perform very well (even better than GPT4o).

I also appreciate the efforts for carrying out various ablative studies regarding the models sizes and the amount of fine-tuning data. I think such statistics would be useful for future works following up on this topic.

**Weaknesses:**

Despite the engineering efforts, the impact of this work seems mostly limited to the domain of web agents and it is unclear what the takeway message is for the broader research community beyond common sense, e.g. fine-tuning with domain-specific data can significantly improve performances and the amount of data needed roughly scales exponentially with the performance improvements.

The paper mentioned that "Due to privacy concerns, we restrict access to our collected data. However, we will release the preprocessing, training, inference code, and a copilot (i.e., browser extension) powered by our agent." which seems a little ambiguous to me. While I understand that collected data might not be shared due to privacy concerns, would the model weights be open-sourced? If the model weights can be shared, I believe that it will have a tremendous impact on the open-source LLM agent community.

---

### Review · Reviewer_SNaP · 2025-04-08

**Summary Of Contributions:**

This paper proposes an approach by fine-tuning open-source LLMs using a large corpus of real-world workflow data (Figure 1), to develop specialized web agents. Through extensive experiments, authors demonstrate that this method not only enhances the web understanding and planning capabilities of LLMs—achieving state-of-the-art performance on various benchmarks—but also enables the development of significantly smaller agent models compared to proprietary LLMs, thereby reducing serving costs. Their empirical findings highlight the critical role of large-scale, high-quality, real-world data in advancing agent development.

**Audience:**

Yes

**Claims And Evidence:**

Yes

**Requested Changes:**

n/a

**Strengths And Weaknesses:**

Strengths:
The authors collected a large-scale dataset of real user workflows from diverse web environments using an AI documentation tool, capturing detailed action sequences with corresponding DOM structures and metadata. Reformatted for next-step prediction, this dataset enables the fine-tuning of open-source LLMs. The result is WorkflowAgent, a family of specialized, single-stage LLM agents that directly predict the next action from the DOM and history, outperforming prior multi-stage agent architectures. The authors state that, due to privacy concerns, access to the collected data will be restricted. However, they will release the preprocessing, training, and inference code, along with a copilot (i.e., browser extension) powered by their agent.

Weaknesses:
From a technical standpoint, this paper offers little novelty. The approach of collecting domain-specific data and fine-tuning an open-source LLM to achieve performance comparable to or better than proprietary LLMs on domain-specific tasks has already been demonstrated numerous times.

---

### Review · Reviewer_um1a · 2025-04-22

**Summary Of Contributions:**

- The main contribution is a huge dataset of production-scale web navigation workflow data collected from over 250 domains with 6 billion tokens. Due to privacy concerns, this dataset is not accessible to reviewers though.
- With the constructed dataset, the authors fine-tune some open-source LLMs and the resulting models beat larger LLMs on challenging web domains including Mind2Web and WebArena.

**Audience:**

Yes

**Broader Impact Concerns:**

The work collects dataset from multiple real-world systems. Without proper ethical framing and safeguards, the benefits of such automation could come at the cost of transparency, privacy, and labor disruption. Maybe the authors can provide a Broader Impact Statement to address the concerns above and clarify the scope of intended use.

**Claims And Evidence:**

Yes

**Requested Changes:**

- Regarding W2.1, It would be nice if the authors present some additional baseline experiments to show the improvement of fine-tuning itself. For example, also give GPT-4o a full page information and do the same trick to automatic scrolling down.
- Some experiments that will help with understanding/addressing the other weaknesses will be helpful, for example ablation studies as in W2.2.
- The authors should carefully check their writing, for example,  Table 8 is wrongly referred to as Figure 8 in the paper.

**Strengths And Weaknesses:**

Strengths
- The paper is generally well-written and easy to follow.
- With the provided dataset, even the WorkflowAgent-Small(7B) model gets significant improvement of results in challenging benchmarks like Mind2Web.

Weaknesses
- W1: In this work, the major contribution is a large dataset and fine-tuning of LLMs on the large dataset. The authors do extensive work, however, the novelty of the work is limited, as the authors simply use existing techniques, for example, fine-tuning LLMs with LoRA.
- W2: unsure about whether some of the experiments are fair
  - W2.1: The model only works on full DOM so that "there is no concept of scrolling in our training dataset". However, the way the authors deal with this is to "determine if the target element is within the visible webpage area" and if not, they will pose some automatic scroll actions. While this is reasonable as a method, its comparison to the baselines may not be fair, because it directly gets rid of the need to imagine, scroll down, and stop when necessary, for a sequential agent, which is inherently difficult.
  - W2.2: Apart from fine-tuning LLMs with collected dataset, there are a lot of tricks applied in evaluations, for example, they aggregate all possible actions and select the one with the highest number of appearances, which does this sampling and selection. However, there are no ablation studies regarding the tricks to show how much they contribute to the work.
  - W2.3: The authors 'only evaluate on tasks whose DOM can fit into the context length', can the authors provide some examples of the tasks that don't fit and are filtered out? Did you run the baselines on the same filtered subset?
  - W2.4: For WebArena - "our agent discovers a workaround that bypasses the need for scrolling through comboboxes". However, I think this is a drawback of the environment itself because it fails to capture the information in the pop-up comboboxes. It is counter-intuitive to type in comboboxes (with knowing the exact keywords ahead of time) and see nothing when the agent click on the comboboxes. Therefore, it may be more fair if the author directly provides this specific type of action to the webarena prompt. However, I'm not very confident in this point, it's just my personal opinion.

---

### Decision · Action_Editor_gZzr · 2025-05-31

**Recommendation:** Reject

**Additional Comments:**

N/A

**Audience:**

No

**Audience Explanation:**

With the proper analysis, and / or data set, model, or source code available, the paper would be of interesting the non-embodied AI agents. However, in its current work the benefit is not clear.

**Claims And Evidence:**

Yes

**Claims Explanation:**

The paper proposes fine-tuning open-source Large Language Models (LLMs) using a large dataset of real-world workflow data collected from over 250 domains, totaling 6 billion tokens. This contrasts with most existing agents that rely on prompting general-purpose, proprietary models not specifically trained for web contexts. The resulting agent, WorkflowAgent, demonstrates significant improvements on the Mind2Web and WebArena. The authors conclude that large-scale, high-quality, real-world data is crucial for agent development and enables the creation of smaller, more cost-effective models that outperform larger proprietary one.

The reviewers raised concerns around the reproducibility of the work and questioning overall value of the presented work. The training dataset and presumably the model cannot be released. The authors promised the code release, however during the review process the code was not available for the review. Given that the authors did not provide the response and clarification, the decisions is based on the presented work, reviewers opinions, and my own reading of the paper.

Overall, the value to the research community and the ability to build upon the presented in its current form work is very limited. It is not surprising that fine-tuning on high-quality dataset improves performance over prompting methods. The paper lacks the insights and know-how of creating such datasets that would make it valuable to the broader research community. Source code availability, and in-depth analysis ( e.g. the dataset's size, domain span influence, workflow selection rationale etc) would have addressed some of these concerns.

**Resubmission Of Major Revision:**

The authors may consider submitting a major revision at a later time.